# EXTRACT: Efficient Policy Learning by Extracting Transferable Robot Skills from Offline Data

**Jesse Zhang[1], Minho Heo[2], Zuxin Liu[3],**
**Erdem Bıyık[1], Joseph J. Lim[2], Yao Liu[4], Rasool Fakoor[4]**
[1]University of Southern California, [2]KAIST, [3]CMU, [4]Amazon Web Services
jessez@usc.edu

**Abstract:** Most reinforcement learning (RL) methods focus on learning optimal policies over low-level action spaces. While these methods can perform well in their training environments, they struggle with learning new tasks. Instead, RL agents acting over useful, temporally extended skills rather than low-level actions can learn new tasks more easily. Prior work in skill-based RL either requires expert supervision to define useful skills, which is hard to scale, or learns a skill-space from offline data with heuristics that limit the adaptability of the skills, making them difficult to transfer. Our approach, EXTRACT, instead utilizes pretrained vision language models to extract a discrete set of semantically meaningful skills from offline data, each of which is parameterized by continuous arguments, *without human supervision*. This skill parameterization allows robots to learn new tasks by only needing to learn *when* to select a specific skill and *how* to modify its arguments for the specific task. We demonstrate through experiments in sparse-reward, image-based, robot manipulation environments, both in simulation and in the real world, that EXTRACT can more quickly learn new tasks than prior works, with major gains in sample efficiency and performance over prior skill-based RL. Our website is at https://jessezhang.net/projects/extract.

## 1 Introduction

Imagine learning to play racquetball as a complete novice. Without prior experience in racket sports, this poses a daunting task that requires learning not only the (1) complex, high-level strategies to control *when* to serve, smash, and return the ball but also (2) *how* to actualize these moves in terms of fine-grained motor control. However, a squash player should have a considerably easier time adjusting to racquetball as they already know how to serve, take shots, and return; they simply need to learn *when* to use these skills and *how* to adjust them for larger racquetball balls. Our paper aims to make use of this intuition to enable efficient learning of new robotics tasks.

In general, humans can learn new tasks quickly—given prior experience—by adjusting existing skills for the new task [1, 2]. Skill-based reinforcement learning (RL) aims to emulate this transfer [3, 4, 5, 6, 7, 8, 9, 10, 11, 12] in learned agents by equipping them with a wide range of skills (i.e., temporally-extended action sequences) that they can call upon for efficient downstream learning. Transferring to new tasks in standard RL, based on low-level environment actions, is challenging because the learned policy becomes more task-specific as it learns to solve its training tasks [13, 14, 15, 16, 17]. In contrast, skill-based RL leverages temporally extended skills that can be both transferred across tasks and yield more informed exploration [6, 18, 19], thereby leading to more effective transfer and learning. However, existing skill-based RL approaches rely on costly human supervision [20, 21, 9, 10] or restrictive skill definitions [22, 6, 8] that limit the expressiveness and adaptability of the skills. Therefore, we ask: how can robots discover *adaptable* skills for efficient transfer learning *without costly human supervision*?

8th Conference on Robot Learning (CoRL 2024), Munich, Germany.

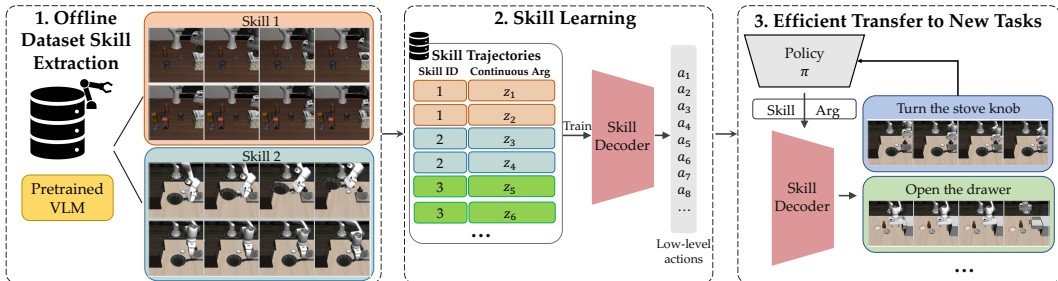

Figure 1: EXTRACT unsupervisedly extracts a discrete set of skills from offline data that can be used for efficient learning of new tasks. (1) EXTRACT first uses VLMs to extract a discrete set of aligned skills from image-action data. (2) EXTRACT then trains a skill decoder to output low-level actions given discrete skill IDs and learned continuous arguments. (3) This decoder helps a skill-based policy efficiently learn new tasks with a simplified action space over skill IDs and arguments.

Calling back to the squash to racquetball transfer example, we humans categorize different racket movements into *discrete skills*—for example, a "forehand swing" is distinct from a "backhand return." These discrete skills can be directly transferred by making minor modifications for racquetball's larger balls and different rackets. This process is akin to that of calling a programmatic API, e.g., `def forehand(x, y)`, where learning to transfer reduces to learning *when* to call discrete functions (e.g., `forehand()` vs `backhand()`) and *how* to execute them (i.e., what their arguments should be). In this paper, we propose a method to accelerate transfer learning by enabling robots to learn, without expert supervision, a discrete set of skills parameterized by input arguments that are useful for downstream tasks (see Figure 1). We assume access to an offline dataset of image-action pairs of trajectories from tasks that are different from the downstream target tasks. Our key insight is aligning skills by extracting *high-level behaviors*, i.e., discrete skills like "forehand swing," from images in the dataset. However, two challenges preclude realizing this insight: (1) how to extract these input-parameterized skills, and (2) how to guide online learning of new tasks with these skills.

To this end, we propose EXTRACT (**Ex**traction of **T**ransferable **R**obot **Act**ion Skills), a framework for extracting discrete, parameterized skills from offline data to guide online learning of new tasks. We first use pre-trained vision-language models (VLMs), trained to align images with language descriptions [23] so that images of similar high-level behaviors are embedded to similar latent embeddings [24], to extract—from our offline data—image embedding differences representing changes in high-level behaviors. Next, we cluster the embeddings in an *unsupervised* manner to form discrete skill clusters that represent high-level skills. To parameterize these skills, we train a *skill decoder* on these clusters, conditioned on the skill ID (e.g., representing a "backhand return") and a learned argument (e.g., indicating velocity), to produce a skill consisting of a temporally extended, variable-length action sequence. Finally, to train a robot for new tasks, we train a skill-based RL policy to act over this skill-space while being guided by skill prior networks, learned from our offline skill data, guiding the policy for (1) *when* to select skills and (2) *what* their arguments should be.

In summary, EXTRACT enables sample-efficient transfer learning for robotic tasks by extracting a meaningful set of skills from offline data for an agent to use for learning new tasks. We first validate that EXTRACT learns a well-clustered set of skills. We then perform experiments across challenging, long-horizon, sparse-reward, image-based robotic manipulation tasks, both in simulation and in the real world on a Panda Franka arm, demonstrating that EXTRACT agents can more quickly transfer skills to new tasks than prior work.

## 2 Related Work

**Defining Skills Manually.** Many works require manual definition of skills, e.g., as pre-defined primitives [4, 25, 26], subskill policies [27, 20, 28], or task sketches [29, 21], making them challenging to scale to arbitrary environments. Closest to ours, Dalal et al. [9] and Nasiriany et al. [10]

hand-define a set of skills parameterized by continuous arguments. But this hand-definition requires expensive human supervision and task-specific, environment-specific, or robot-specific fine-tuning. In contrast, EXTRACT *automatically* learns skills from offline data, which is much more scalable to enable learning multiple downstream tasks. We demonstrate in Section 5 that, given sufficient data coverage, skills extracted from data can transfer as effectively as hand-defined skills.

**Unsupervised Skill Learning.** A large body of prior work discovers skills in an unsupervised manner to accelerate learning new tasks. Some approaches use heuristics to extract skills from offline data, like defining skills as randomly sampled trajectories [30, 31, 32, 33, 34, 6, 8, 7, 35]. While these approaches have demonstrated that randomly sampled skill sequences can accelerate downstream learning, EXTRACT instead uses visual embeddings from VLMs to combine sequences performing similar behaviors into the same skill while allowing for intra-skill variation through their arguments. We show in Section 5 that our skill parameterization allows for more efficient online learning than randomly assigned skills. Moreover, Wan et al. [36] also learns skills via clustering visual features; however, in addition to major differences in methodology, they focus on *imitation* learning—requiring significant algorithmic changes to facilitate learning new tasks online [37, 38, 39]. Instead, we directly focus on online *reinforcement* learning of new tasks.

Another line of work aims to discover skills for tasks without offline data. Some learn skills while simultaneously attempting to solve the task [3, 40, 5, 41, 19, 11]. However, learning the skills and using them simultaneously is challenging, especially without dense reward supervision. Finally, some prior works construct unsupervised objectives, typically based on entropy maximization, to learn task-agnostic behaviors [42, 43, 44, 45, 46]. However, these entropy maximization objectives lead to learning a large set of skills, most of which form random behaviors unsuitable for any meaningful downstream task. Thus, using them to learn long-horizon, sparse-reward tasks is difficult. We focus on first extracting skills from demonstration data, assumed to have meaningful behaviors to learn from, for online learning of unseen, sparse-reward tasks.

## 3 Preliminaries

**Problem Formulation.** We assume access to an offline dataset of trajectories $\mathcal{D} = \{\tau_1, \tau_2, ...\}$ where each trajectory consists of ordered image observation and action tuples, $\tau_i = [(s_1, a_1), (s_2, a_2), ...]$. The downstream transfer learning problem is formulated as a Markov Decision Process in which we want to learn a policy $\pi$ to maximize downstream rewards. We note that the offline dataset $\mathcal{D}$ does not contain trajectories from downstream task(s); we assume that the state space $\mathcal{S}$ has the same dimensions and that actions in $\mathcal{D}$ can be used to solve downstream tasks.

**SPiRL.** In order to extract skills from offline data and use these skills for a new policy, we build on top of a previous skill-based RL method, namely SPiRL [6]. SPiRL focused on learning skills defined by randomly sampled, fixed-length action sequences. We briefly summarize SPiRL here: Given $H$-length sequences of consecutive actions from $\mathcal{D}$: $\bar{a} = a_1, ..., a_H$, SPiRL learns (1) a generative **skill decoder** model, $p_a(\bar{a} \mid z)$, which decodes learned, latent skills $z$ encoded by a **skill encoder** $q(z \mid \bar{a})$ into environment action sequences $\bar{a}$, and (2) a state-conditioned skill **prior** $p_z(z \mid s)$ that predicts which latent skills $z$ are likely to be useful at state $s$. To learn a new task, SPiRL trains a skill-based policy $\pi(z \mid s)$, whose outputs $z$ are skills decoded by $p_a(\bar{a} \mid z)$ into low-level environment actions. The objective of policy learning is to maximize returns under $\pi(z \mid s)$ with a KL divergence constraint to regularize $\pi$ against the prior $p_z(z \mid s)$.

## 4 Method

EXTRACT aims to discover a discrete skill library from an offline dataset that can be modulated through input arguments for learning new tasks efficiently. EXTRACT operates in three stages: (1) an offline skill *extraction* stage, (2) an offline skill *learning* phase in which we train a decoder model to reproduce action sequences given a skill choice and its arguments, and finally (3) the online RL stage for training an agent to utilize these skills for new tasks. See Figure 2 for a detailed overview.

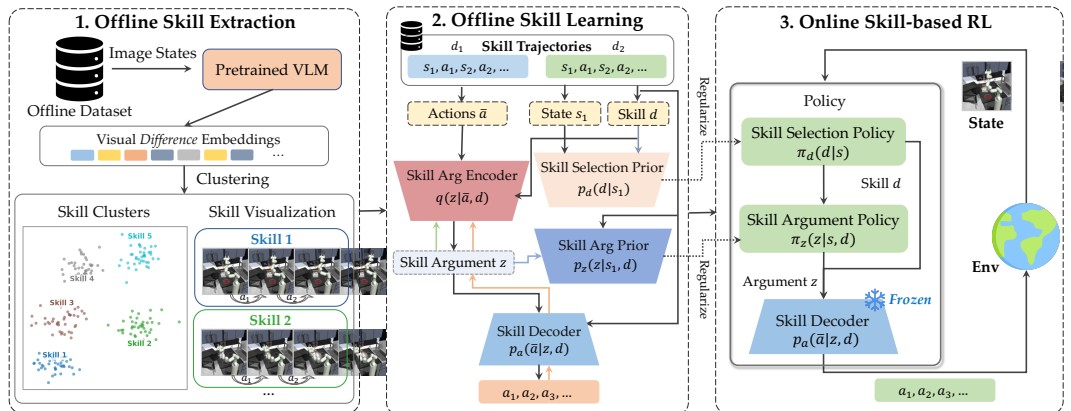

Figure 2: EXTRACT consists of three phases. **(1) Skill Extraction**: We extract a discrete set of skills from offline data by clustering together visual VLM difference embeddings representing high-level behaviors. **(2) Skill Learning**: We train a skill decoder model, $p_a(\bar{a} \mid z, d)$, to output variable-length action sequences conditioned on a skill ID $d$ and a learned continuous argument $z$. The argument $z$ is learned by training $p_a(\bar{a} \mid z, d)$ with a VAE reconstruction objective from action sequences encoded by a skill encoder, $q(z \mid \bar{a}, d)$. We additionally train a skill selection prior and skill argument prior $p_d(d \mid s)$, $p_z(z \mid s, d)$ to predict which skills $d$ and their arguments $z$ are useful for a given state $s$. Colorful arrows indicate gradients from reconstruction, argument prior, selection prior, and VAE losses. **(3) Online RL**: To learn a new task, we train a skill selection and skill argument policy with RL while regularizing them with the skill selection and skill argument priors.

## 4.1 Offline Skill Extraction

**Feature extraction.** We leverage vision-language models (VLMs), trained to align large corpora of images with natural language descriptions [23, 47, 48, 49], to extract high-level features used to label skills. Although our approach does not require the use of language, we utilize VLMs because, as VLMs were trained to align images with *language*, VLM image embeddings represent a *semantically aligned* embedding space. However, one main issue precludes the naïve application of VLMs in robotics. In particular, VLMs do not inherently account for object variations or robot arm starting positions across images [50, 24, 51, 52]. But in robot manipulation, high-level behaviors should be characterized by *changes* in arm and object positions across a trajectory—picking up a cup should be considered the same skill regardless of if the cup is to the robot's left or right. Our initial experiments of using the embeddings directly resulted in skills specific to one type of environment layout or object. Therefore, to capture high-level behaviors, we use trajectory-level *embedding differences* by taking the difference of each VLM image embedding with the first one in the trajectory:[1]

$$e_t = \text{VLM}(s_t) - \text{VLM}(s_1). \tag{1}$$

**Skill label assignment.** After creating embeddings $e_t$ for each image $s_t$, we assign skill labels in an *unsupervised* manner based on these features. Inspired by classical algorithms from *speaker diarization*, a long-studied problem in speech processing where the objective is to assign a "speaker label" to each speech timestep [53], we first perform unsupervised clustering with K-means on the entire dataset of embedding differences $e_i$ to assign per-timestep skill labels (the label is the cluster ID), then we smooth out the label assignments with a simple median filter run along the trajectory sequence to reduce the frequency of single or few-timestep label assignments. See Figure 3 for a visual demonstration of this process.

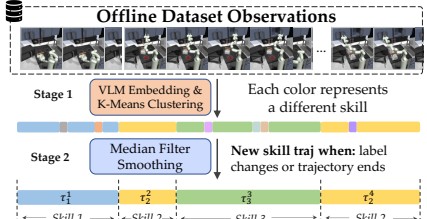

Figure 3: Skill label assignment consists of (1) using the VLM embedding differences for clustering, then (2) applying a median filter over the labels to smooth out noisy assignments.

---

[1]To ensure that each timestep has an embedding, we assign embedding $e_1$ to be identical to $e_2$.

In summary, we first extract observation embedding difference features with a VLM and then perform unsupervised K-means clustering to obtain skill labels for each trajectory timestep. This forms the skill-labeled dataset $\mathcal{D}_d = \{\tau_d^1, \tau_d^2, ...\}$, where each $\tau_d$ is a trajectory of sequential $(s, a)$ tuples that all belong to one skill $d$. Next, we perform skill learning on $\mathcal{D}_d$.

## 4.2 Offline Skill Learning

We aim to learn a *discrete* set of skills, parameterized by *continuous* arguments, similar to a functional API over skills (see Figure 2 middle). Therefore, we train a generative **skill decoder** $p_a(\bar{a} \mid z, d)$ to convert a discrete skill choice $d$ and a continuous argument for that skill, $z$, into an action sequence. As alluded to in Section 3, we build upon SPiRL by Pertsch et al. [6]. However, they train their decoder to decode fixed-length action sequences from a single continuous latent $z$. In contrast, we automatically extract a set of variable-length skill trajectories with labels denoted $d$ and parameterize each skill by a learned, continuous latent argument $z$.[2]

We train an autoregressive VAE [54] consisting of the following learned neural network components: a skill argument **encoder** $q(z \mid \bar{a}, d)$ mapping to a continuous latent $z$ conditioned on a discrete skill choice $d$ and an action sequence $\bar{a}$, and an autoregressive skill **decoder** $p_a(\bar{a} \mid z, d)$ conditioned on the latent $z$ and the discrete skill choice $d$.[3] Because the action sequence $\bar{a}$ can be of various lengths, the decoder also learns to produce a continuous value $l$ at each autoregressive timestep representing the proportion of the skill completed at the current action. This variable is used during online RL to stop the execution of the skill when $l$ equals 1 (see Appendix B.1 for further details).

Recall that SPiRL also trains a skill prior network $p_z(z \mid s)$ that predicts which $z$ is useful for an observation $s$; this prior is used to guide a high-level policy toward selecting reasonable $z$ while performing RL. In contrast with SPiRL where $z$ uniquely represents a skill, we train *two* prior networks, one to guide the selection of the skill $d$, $p_d(d \mid s)$, and one to guide the selection of its argument $z$ given $d$, $p_z(z \mid s, d)$. These are trained with the observation from the first timestep of the sampled trajectory, $s_1$, to be able to guide a skill-based policy during online RL in choosing $d$ and $z$. Our full objective for training this VAE is to maximize the following:

$$\mathbb{E}_{\substack{\bar{a}, d, s_1 \sim \mathcal{D}_d \\ z \sim q(\cdot \mid \bar{a}, d)}} \left[ \underbrace{\left[ \sum_{t=1}^{|\bar{a}|} \log p_a(a_t, l \mid z, d) \right]}_{\text{action rec. + progress pred.}} + \underbrace{\beta \, \text{KL}\left( q(z \mid \bar{a}, d) \parallel N(0, I) \right)}_{\text{VAE encoder KL regularization}} + \underbrace{\log p_d(d \mid s_1)}_{\text{discrete skill prior}} + \underbrace{\log p_z(\textbf{sg}(z) \mid s_1, d)}_{\text{continuous arg. prior}} \right], \quad (2)$$

where the stop-gradient $\textbf{sg}(\cdot)$ prevents prior losses from influencing the encoder and $z$ is sampled from the encoder $q(z \mid \bar{a}, d)$. The first two terms are the $\beta$-VAE objective [55]; the last two train priors to predict the correct skill $d$ and continuous argument $z$ given $s_1$.

**Additional fine-tuning.** On extremely challenging transfer scenarios, demonstrations may still be needed to warm-start reinforcement learning [56]. EXTRACT can also flexibly be applied to this setting by using the same K-means clustering model from Section 4.1, which was trained to cluster $\mathcal{D}_d$, to assign skill labels to an additional, smaller demonstration dataset. After pre-training on $\mathcal{D}_d$, we then fine-tune the entire model on that labeled demonstration dataset before performing RL.

## 4.3 Online Skill-Based Reinforcement Learning

Finally, we describe how we perform RL for new tasks by training a skill-based policy to select skills and their arguments to solve new tasks. See Figure 2, right, for an overview of online RL.

**Policy parameterization.** After pre-training the decoder $p_a(\bar{a} \mid z, d)$, we treat it as a frozen lower-level policy that a learned skill-based policy can use to interact with a new task. Specifically, we train a skill-based policy $\pi(d, z \mid s)$ to output a $(d, z)$ tuple representing a discrete skill choice and its continuous argument. We parameterize this policy as a product of two policies: $\pi(d, z \mid s) = \pi_d(d \mid s)\pi_z(z \mid s, d)$ so that each component of $\pi(d, z \mid s)$ can be regularized with our pre-trained

---

[2]To simplify notation, we use $z$ for both our method and SPiRL. However, it is important to note that $z$ uniquely determines the skill in SPiRL, while $z$ denotes a continuous latent argument in our method.

[3]$p_a(\bar{a} \mid z, d)$ can also be state-conditioned. We opt not to for better *transfer* to new tasks with unseen states.

priors $p_d(d \mid s)$ and $p_z(z \mid s, d)$. Intuitively, this parameterization separates decision-making into *what* skill to use and *how* to use it. The complete factorization of the skill-based policy follows:

$$\pi(a \mid s) = \underbrace{p_a(\bar{a} \mid z, d)}_{\text{skill decoder}} \cdot \pi(d, z \mid s) = \underbrace{p_a(\bar{a} \mid z, d)}_{\text{skill decoder}} \cdot \underbrace{\pi_d(d \mid s) \cdot \pi_z(z \mid s, d)}_{\text{learned skill-based policy}}. \qquad (3)$$

**Policy learning.** We can train the skill-based policy with online data collection using any entropy-regularized RL algorithm, such as SAC [57] or RLPD [58], where we regularize against the skill priors instead of against a max-entropy uniform prior. Because we have factorized $\pi(d, z \mid s)$ into two separate policies, we can easily regularize each with the priors trained in Section 4.2. The training objective for the policy with SAC is to maximize over $\pi_d, \pi_z$:

$$\mathbb{E}_{\substack{s, d \sim \pi_d(. \mid s) \\ z \sim \pi_z(. \mid s, d)}} \Big[ Q(s, z, d) - \alpha_z \underbrace{\mathrm{KL}(\pi_z(z \mid s, d) \parallel p_z(\cdot \mid s, d))}_{\text{skill argument guidance}} - \alpha_d \underbrace{\mathrm{KL}(\pi_d(d \mid s) \parallel p_d(\cdot \mid s))}_{\text{skill choice guidance}} \Big], \quad (4)$$

where $\alpha_z$ and $\alpha_d$ control the prior regularization weights. The critic objective is also correspondingly modified (see Appendix Algorithm 4). Despite the hierarchical architecture, this objective is stable to train as the lower-level skill decoder is frozen and the priors regularize the high-level policy.

In summary, EXTRACT first extracts a set of discrete skills from offline image-action data (Section 4.1), then trains an action decoder to take low-level actions in the environment conditioned on a discrete skill and continuous latent (Section 4.2), and finally performs prior-guided reinforcement learning over these skills online in the target environment to learn new tasks (Section 4.3). See Algorithm 1 (appendix) for the pseudocode and Appendix B.1 for additional implementation details.

## 5 Experiments

Our experiments investigate the following questions: (1) Does EXTRACT discover meaningful, well-aligned skills from offline data? (2) Do EXTRACT-acquired skills help robots learn new tasks? (3) What components of EXTRACT are important in enabling transfer?

### 5.1 Experimental Setup

We evaluate EXTRACT on two long-horizon, continuous-control, robotic manipulation domains: Franka Kitchen [59] and LIBERO [60]. and the real-world FurnitureBench [61]. All environments use image observations and sparse rewards. For both Franka Kitchen and LIBERO, our method **EXTRACT** uses the R3M VLM [47] and K-means with $K = 8$ for offline skill extraction (Section 4.1). In FurnitureBench, $K = 6$. We list specific details below; see Appendix B.3 for more.

**Franka Kitchen:** This environment, originally from Gupta et al. [22], Fu et al. [59] contains a Franka Panda arm operating in a kitchen environment. Similarly to Pertsch et al. [6], we test transfer learning of a sequence of 4 subtasks never performed in sequence in the dataset. Agents are given a reward of 1 for completing each subtask. 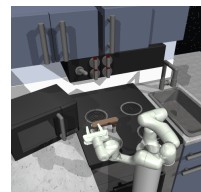

**LIBERO:** LIBERO [60] consists of a Franka Panda arm interacting with many objects and drawers. We test transfer to four task suites, LIBERO-{`Object`, `Spatial`, `Goal`, `10`} consisting of 10 unseen environments/tasks each, spanning various transfer scenarios (40 total tasks). LIBERO tasks are language conditioned (e.g., "turn on the stove and put the moka pot on it"); for pre-training and RL, we condition all methods on the language instruction. Due to 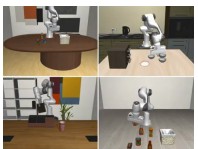 LIBERO's difficulty [62], for all pre-trained methods, we first fine-tune to a provided additional target task dataset with 50 demos per task before performing RL. During RL, we fine-tune on all tasks within each suite simultaneously. To the best of our knowledge, we are the first to report successful RL results on LIBERO tasks.

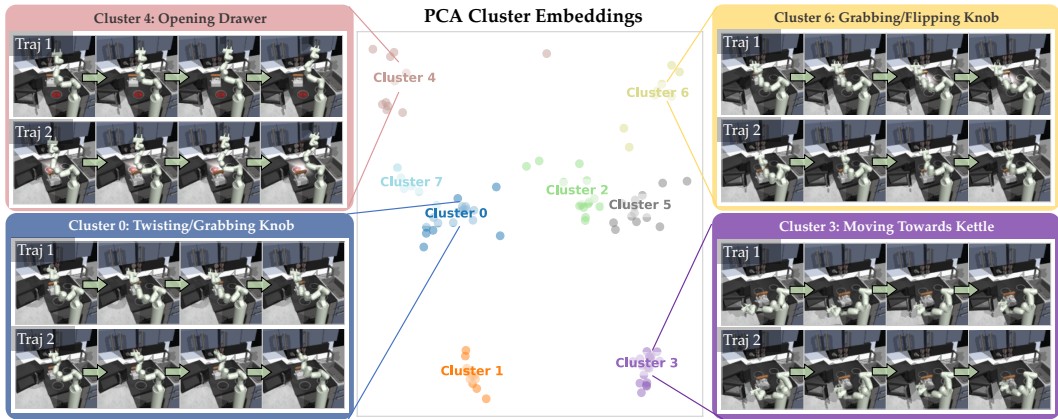

Figure 4: 100 randomly sampled trajectories from the Franka Kitchen dataset after being clustered into skills and visualized in 2D (originally 2048) with PCA. Even in 2 dimensions, clusters can be clearly distinguished. We visualize 2 randomly sampled skills in each cluster, demonstrating that our skill assignment mechanism successfully aligns trajectories performing similar high-level behaviors.

**FurnitureBench:** FurnitureBench [61] tests an agent's ability to assemble real-world furniture with a Franka Panda arm. We pre-train on `one-leg` assembly data without initial object placement randomness and test real-world RL transfer to the same task with ± 5cm of initial object and end-effector position randomness, plus 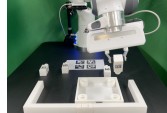 ± 15 degrees of end-effector angle randomization. We use RLPD [58], a sample-efficient actor-critic RL algorithm, for more efficient real-world training. RL is run for 100 training trajectories after pre-training on 500 demonstration trajectories.

**Baselines and Comparisons.** We compare: (1) an oracle (**RAPS** [9]), which is given *ground truth* discrete skills, with continuous input arguments, designed by humans specifically for Franka Kitchen; (2) methods that pre-train with the same data—namely **SPiRL** [6] which extracts sequences of fixed-length random action trajectories as skills, EXTRACT-**UVD** which replaces our discrete skill extraction with UVD's VLM-based mechanism [63], and **BC**, behavior cloning using the same offline data but no temporally extended skills; and (3) **SAC** [57], i.e., RL without any offline data. See Appendix B for implementation details. Sim results include standard deviations over 5 seeds.

## 5.2 Offline Skill Extraction

We first test EXTRACT's ability to discover meaningful, well-aligned skills during skill extraction. In Figure 4, we plot K-means ($K = 8$) skill assignments in Franka Kitchen. We project VLM embedding differences down to 2-D with PCA for visualization. These skill assignments demonstrate that unsupervised clustering of VLM embedding differences can create distinctly separable clustering assignments. For example, skill 4 (Figure 4, top left) demonstrates a cabinet opening behavior. See additional visualizations for all environments in Appendix C.1. We also analyze quantitative clustering statistics in Appendix C.2. Next, let's see how these skills help with learning new tasks.

## 5.3 Online Reinforcement Learning of New Tasks

**Simulated Envs.** We investigate the ability of all methods to transfer to new tasks in simulation in Figure 5. In Kitchen, EXTRACT matches the oracle performance while being **10x** more sample-efficient than SPiRL, with SPiRL needing 3M timesteps to reach EXTRACT's performance at 300k. In all LIBERO suites, EXTRACT performs best in either sample efficiency or final performance due to its discrete-continuous skill separation enabling easier downstream RL; it outperforms SPiRL and EXTRACT-UVD the most in LIBERO-10, the suite with the longest-horizon tasks. EXTRACT-UVD is unstable in Franka Kitchen (see Appendix C.4 for analysis) and generally performs worse as UVD's skill extraction mechanism does not perform our discrete skill clustering. Meanwhile, SAC and BC perform poorly, indicating our tasks are difficult to solve with standard RL or without skills.

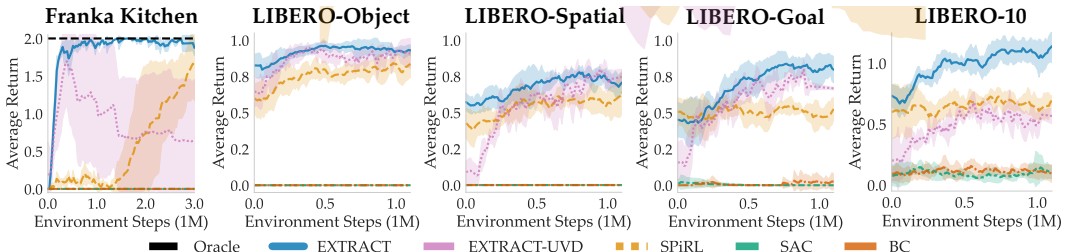

Figure 5: EXTRACT outperforms SPiRL and EXTRACT-UVD in RL across all comparisons, demonstrating the advantages of our clustered skill-space. SAC and BC struggle, demonstrating the need for skill-based RL. In LIBERO-{Object, Spatial, Goal}, return is success rate.

Our method outperforms others due to its semantically aligned, discrete skill-space. For example, to open drawers, EXTRACT's policy only needs to learn a single discrete drawer-opening skill when the gripper is near any drawer. In contrast, SPiRL requires memorizing and distinguishing continuous skills for each specific drawer-opening behavior. Additionally, EXTRACT allows easier exploration later in the task, enabling the policy to reuse the same skill for other drawers. For more details, see Appendix E. Next, we conduct an ablation study on EXTRACT's components.

**Real world.** Finally, we assess EXTRACT's real-world performance on FurnitureBench for `one-leg` assembly in Table 1. We report the average completed subtask (20 trials) out of a maximum of 5. EXTRACT outperforms SPiRL both before and after 100 episodes of real-world RL fine-tuning, showing effective skill transfer. Overall, EXTRACT excels across 42 tasks and 3 domains, outperforming other skill-based RL, BC, and online RL methods, both in simulation and on robots.

Table 1: Furniture RL.

| Method | Start | End |
|---|---|---|
| SPiRL | 1.35 | 1.55 |
| EXTRACT | **1.90** | **2.50** |

### 5.4 EXTRACT RL Ablation Studies

**VLMs.** We first ablate the use of VLMs from selecting features for clustering. Therefore, we compare against **Action**, where skill labels are generated by clustering robot *action differences*. We also compare against **State** where skills are labeled by clustering ground truth state differences (e.g., robot joints, states of all objects). State represents an oracle scenario as ground truth states of all relevant objects are difficult to obtain in the real world. We plot results in Franka Kitchen in Figure 6. EXTRACT with VLM-extracted skills performs best, as both ground truth state and raw environment action differences can be difficult to directly obtain *high-level*, semantically meaningful skills from. For ablations against pure proprioception and CLIP [23], see Appendix C.3.

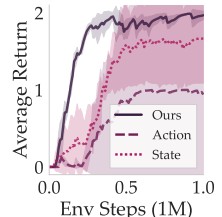

Figure 6: Embedding ablations.

**Number of Clusters.** Finally, we ablate the number of K-means clusters. Too few or too many clusters can affect RL by balancing the ease of selecting the correct discrete skill against the complexity of choosing the right continuous argument. In Figure 7, we show average returns at 1M timesteps for EXTRACT in Kitchen with $K = 3, 5, 8, 15$. Performance remains stable, with a drop only at $K = 15$, indicating that EXTRACT is robust to variations in the number of discovered discrete skills.

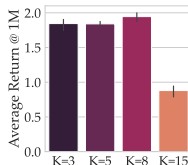

Figure 7: Kitchen $K$ ablations.

## 6 Discussion

We presented EXTRACT, a method for enabling efficient agent transfer learning by extracting a discrete set of input-argument parameterized skills from offline data for a robot to use in new tasks. Compared to standard RL, our method operates over temporally extended skills rather than low-level environment actions, providing greater flexibility and transferability to new tasks, as demonstrated by our comprehensive experiments. Our experiments demonstrated that EXTRACT performs well across 41 total tasks across 2 robot manipulation domains. We discuss limitations in Appendix F.

## Acknowledgments

Most of this work was performed while Jesse Zhang and Zuxin Liu were interns at Amazon Web Services. After the internships, this work was supported by a USC Viterbi Fellowship, compute infrastructure from AWS, Institute of Information & Communications Technology Planning & Evaluation (IITP) grants (No.RS-2019-II190075, Artificial Intelligence Graduate School Program, KAIST; No.RS-2022-II220984, Development of Artificial Intelligence Technology for Personalized Plug-and-Play Explanation and Verification of Explanation), a National Research Foundation of Korea (NRF) grant (NRF-2021H1D3A2A03103683, Brain Pool Research Program) funded by the Korean government (MSIT), Electronics and Telecommunications Research Institute (ETRI) grant funded by the Korean government foundation [24ZB1200, Research of Human-centered autonomous intelligence system original technology], and Samsung Electronics Co., Ltd (IO220816-02015-01). Finally, we thank Laura Smith and Sidhant Kaushik for their valuable feedback on early versions of the paper draft.

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

# A  Full Algorithm

---

**Algorithm 1** EXTRACT Algorithm, Section 4.

---

**Require:** Dataset $\mathcal{D}$, VLM, Target MDP $\mathcal{M}$, Optional target task fine-tuning dataset $\mathcal{D}_{\mathcal{M}}$
1: $\mathcal{D}_d, CM \leftarrow$ OFFLINESKILLEXTRACTION($\mathcal{D}$, VLM)     ▷ Get discrete skill labels and clustering model, Algorithm 2
2: Init $q(z \mid \bar{a}, d), p_a(\bar{a} \mid z, d), p_d(d \mid s), p_z(z \mid s, d)$     ▷ Skill argument encoder, skill decoder, discrete skill prior, continuous argument prior
3: $q, p_a, p_d, p_z \leftarrow$ OFFLINESKILLLEARNING($\mathcal{D}_d, q, p_a, p_d, p_z$)     ▷ Learn skills offline, Algorithm 3
4: **if** $\mathcal{D}_{\mathcal{M}}$ exists **then**
5:     $\mathcal{D}_{\mathcal{M},d} \leftarrow$ Assign skills to $\mathcal{D}_{\mathcal{M}}$ with existing clustering model $CM$
6:     $q, p_a, p_d, p_z \leftarrow$ OFFLINESKILLLEARNING($\mathcal{D}_{\mathcal{M},d}, q, p_a, p_d, p_z$)     ▷ Optionally fine-tune on target task $\mathcal{M}$
7: **end if**
8: SKILLBASEDONLINERL($\mathcal{M}, p_a, p_d, p_z$)     ▷ RL on target task $\mathcal{M}$, Algorithm 4

---

**Algorithm 2** Offline Skill Extraction, Section 4.1.

---

1: **procedure** OFFLINESKILLEXTRACTION($\mathcal{D}$, VLM)
2:     EMBEDS $\leftarrow$ []     ▷ Init VLM embedding differences
3:     **for** trajectory $\tau = [(s_1, a_1), ..., (s_T, a_T)]$ in $\mathcal{D}$ **do**
4:         **for** $(s_i, a_i)$ in $\tau$ **do**
5:             $e_i = $ VLM($s_i$) $-$ VLM($s_1$)     ▷ Embedding differences, Equation (1)
6:             EMBEDS.APPEND($e_i$)
7:         **end for**
8:     **end for**
9:     $CM \leftarrow$ Init (K-Means) clustering model
10:    LABELS $\leftarrow CM$(EMBEDS)     ▷ Run unsupervised clustering to get cluster labels
11:    $\mathcal{D}_d \leftarrow \{\}$     ▷ Init skill labeled dataset
12:    **for** trajectory $\tau = [(s_1, a_1), ..., (s_T, a_T)]$ in $\mathcal{D}$ **do**
13:        $d_1, ..., d_T \leftarrow$ Get labels from LABELS
14:        $d_1, ..., d_T \leftarrow$ MEDIANFILTER($d_1, ..., d_T$)     ▷ Smooth out labels, see Appendix B.1
15:        $\mathcal{D}_d \leftarrow \mathcal{D}_d \cup [(s_1, a_1, d_1), ..., (s_T, a_T, d_T)]$
16:    **end for**
17:    **return** $\mathcal{D}_d, CM$
18: **end procedure**

---

**Algorithm 3** Offline Skill Learning, Section 4.2.

---

1: **procedure** OFFLINESKILLLEARNING($\mathcal{D}, q, p_a, p_d, p_z$)
2:     **while** not converged **do**
3:         Sample $\tau_d$ from $\mathcal{D}_d$
4:         Train $q, p_a, p_d, p_z$ with Equation (2)
5:     **end while**
6:     **return** $q, p_a, p_d, p_z$
7: **end procedure**

---

We present the full EXTRACT pseudocode in Algorithm 1. Algorithm 2 details offline skill extraction using a VLM, Algorithm 3 details the offline skill learning procedure, and Algorithm 4 details how to perform online skill-based RL on downstream tasks using Soft Actor-Critic (SAC). Note that any entropy-regularized algorithm can be used here with similar modifications, not just SAC. Differences from SAC during online RL are highlighted in red. For further implementation details and hyperparameters of EXTRACT, see Appendix B.1.

**Algorithm 4** Skill-Based Online RL (with SAC [57]), Section 4.3. Red marks policy and critic loss differences against SAC.

---

1: **procedure** SKILLBASEDONLINERL($\mathcal{M}, p_a(\bar{a} \mid z, d), p_d(d \mid s), p_z(z \mid s, d)$)  ▷ Section 4.3
2:    Freeze $p_a(\bar{a} \mid z, d), p_d, p_z$ weights
3:    $\pi_d(d \mid s) \leftarrow p_d(d \mid s)$  ▷ Init $\pi_d$ as discrete skill prior $p_d$
4:    $\pi_z(z \mid s, d) \leftarrow p_z(z \mid s, d)$  ▷ Init $\pi_z$ as cont. argument prior $p_z$
5:    $B \leftarrow \{\}$  ▷ Init buffer B
6:    **for** each rollout **do**
7:        $l \leftarrow 0$
8:        $d_t \sim \pi_d(d \mid s_t)$  ▷ Sample discrete skill
9:        $z_t \sim \pi_z(z \mid s, d_t)$  ▷ Sample continuous argument for skill
10:       $a_1, ..., a_L, l_1....l_L \leftarrow \bar{a} \sim p_a(\bar{a} \mid z_t, d_t)$  ▷ Sample action sequence $a_1, ..., a_L$ and
  progress predictions $l_1, ..., l_l$ up to max sequence length $L$, see Appendix B.1.2.
11:       **for** $a$ in $a_1., ..., a_L$ or until $l \geq 1$ **do**
12:           Execute actions in $\mathcal{M}$, accumulating reward sum $\tilde{r}_t$
13:       **end for**
14:       $B \leftarrow B \cup \{s_t, z_t, \tilde{r}_t, s_{t'}\}$  ▷ Add sample to buffer
15:       $(s, z, \tilde{r}, s') \sim B$  ▷ Sample from $B$
16:       $\pi_d, \pi_z \leftarrow \max\limits_{\pi_d, \pi_z} Q(s, z, d)$
17:           $\textcolor{red}{-\alpha_z \mathrm{KL}(\pi_z(z \mid s, d) \parallel p_z(\cdot \mid s, d))}$
18:           $\textcolor{red}{-\alpha_d \mathrm{KL}(\pi_d(d \mid s) \parallel p_d(\cdot \mid s))}$  ▷ Update policies, Equation (4)
19:       $Q \leftarrow \min_Q Q(s, z, d) = r(s, z, d) + \gamma Q(s', z', d')$
20:           $\textcolor{red}{-\alpha_z D_{KL}(\pi(z \mid s, d) \parallel p_z(\cdot \mid s, d))}$
21:           $\textcolor{red}{-\alpha_d D_{KL}(\pi(d \mid s) \parallel p_d(\cdot \mid s))}$  ▷ Update critic
22:    **end for**
23: **end procedure**

---

# B   Experiment and Implementation Details

In this section, we list implementation details for EXTRACT (Appendix B.1), the specific environment setups (Appendix B.3), and details for how we implemented baselines (Appendix B.2).

## B.1   EXTRACT Implementation Details

EXTRACT implementation details follow in the same order as each method subsection was presented in the main paper in Section 4.

### B.1.1   Offline Skill Extraction

We first extract skills from a dataset $\mathcal{D}$ using a VLM by clustering VLM embedding differences of image observations in $\mathcal{D}$ (see pseudocode in Algorithm 2).

**Clustering.**   We use K-means for the clustering algorithm as it is performant, time-efficient, and can be easily utilized in a batched manner if all of the embeddings are too large to fit in memory at once.[4] When extracting skills from the offline dataset $\mathcal{D}$, we utilize K-means clustering on VLM embedding differences with $K = 8$ in Franka Kitchen and LIBERO, as we found $K = 8$ to produce the most visually pleasing clustering assignments in Franka Kitchen and we directly adapted the Franka Kitchen hyperparameters to LIBERO to avoid too much environment-specific tuning. In FurnitureBench, we found $K = 6$ to produce the most visually distinguishable clustering assignments.

---

[4]We did perform preliminary experiments early on with DBSCAN, which doesn't require presetting the number of clusters. However, DBSCAN requires an $\epsilon$ parameter which we found to greatly affect the skill clustering results on our datasets, with some values of $\epsilon$ resulting in very poorly clustered skills.

**Median Filtering.** After performing K-means, we utilize a standard median filter, as is commonly performed in classical speaker diarization [53], to smooth out any possibly noisy assignments (see Figure 3). Specifically, we use the Scipy `scipy.signal.medfilt(kernel_size=7)` [64] filter for all environments. This corresponds to a median filter with window size 7 that slides over each trajectory's labels and assigns the median label within that window to all 7 elements. Empirically, we found that this increased the average length of skills as it reduced the occurrence of short, noisy assignments.

### B.1.2 Offline Skill Learning

Here, we train a VAE consisting of skill argument encoder $q(z \mid \bar{a}, d)$, skill decoder $p_a(\bar{a} \mid z, d)$, discrete skill prior $p_d(d \mid s)$, and continuous skill argument prior $p_z(z \mid s, d)$ (see pseudocode in Algorithm 3).

**Model architectures.** We closely follow SPiRL's model architecture implementations [6] as we build upon SPiRL. The encoder $q(z \mid \bar{a}, d)$ and decoder $p_a(\bar{a} \mid z, d)$ are implemented with recurrent neural networks. The skill priors are both standard multi-layer perceptrons. The skill argument space $z$ has 5 dimensions. In Kitchen and LIBERO, our $\beta$ for the $\beta$-VAE KL regularization term in Equation (2) is 0.001.

**Skill progress predictor.** During training, for GPU memory reasons, we sample skill trajectories with a maximum length as is common when training autoregressive models. In Franka Kitchen, this is heuristically set to 30 based on reconstruction losses and in LIBERO, this is set to 40. In FurnitureBench, this is set to 30. If a skill trajectory is longer than this maximum length, we simply sample a random contiguous sequence of the maximum length within the trajectory. To ensure that predicted action sequences stay in-distribution with what was seen during training, we also use these maximum lengths as maximum skill lengths during online RL; e.g., if a skill runs for 30 timesteps in Franka Kitchen without stopping, we simply resample the next skill (see Line 10 of Algorithm 4).

As discussed in Section 4.2, given the variable lengths of action sequences $\bar{a}$, the decoder $p_a(\bar{a} \mid z, d)$ is trained to generate a continuous skill progress prediction value $l$ at each timestep. This value represents the proportion of the skill completed at the current time. During online policy rollouts, the execution of the skill is halted when $l$ reaches 1. To learn this progress prediction value, we formulate it as follows: when creating labels for such a sequence, we assign a label to each time step, denoted as $y_t$, based on its position in the sequence. Specifically, $y_t$ is set to $\frac{t}{N}$ for each time step $t$, where $N$ represents the sequence length. To train the model for this function, we use the standard mean-squared error loss. This ensures that the model learns to predict the end of an action sequence while also ensuring that it receives dense, per-timestep supervision while training function.

**Additional target task fine-tuning.** Optionally, for very difficult tasks, some target-task demonstrations may be needed [56, 62, 60]. We perform additional target task fine-tuning in LIBERO [60] and FurnitureBench [61]. We use the learned clustering model that was trained to cluster the original dataset $\mathcal{D}$ to directly assign labels to the task-specific dataset $\mathcal{D}_\mathcal{M}$ without updating the clustering algorithm parameters (see Algorithm 1 Line 5). Then, we fine-tune the entire model, $q, p_a, p_d, p_z$, with the same objective in Equation (2) on the labeled target-task dataset $\mathcal{D}_{\mathcal{M},d}$.

### B.1.3 Skill-Based Online RL

For online RL, we utilize the pre-trained skill decoder $p_a(\bar{a} \mid z, d)$, and the skill priors $p_d(d \mid s), p_z(z \mid s, d)$ for skill-based policy learning (see Algorithm 4).

**Policy learning.** Our policy skill-based policy $\pi(d, z \mid s)$ is parameterized as a product of a discrete skill selection policy $\pi_d(d \mid s)$ and a continuous argument selection policy $\pi_z(z \mid s, d)$ (see Equation (4)). To train with actor-critic RL, we sum over the policy losses in each discrete skill dimension weighted by the probability of that skill, similar to discrete SAC loss proposed by

Christodoulou [65]:

$$\sum_d \pi_d(d \mid s)\Big(Q(s,z,d) - \alpha_z \text{KL}(\pi_z(z \mid s,d) \parallel p_z(\cdot \mid s,d)) - \alpha_d \text{KL}(\pi_d(d \mid s) \parallel p_d(\cdot \mid s))\Big). \quad (5)$$

Meanwhile, critic losses are computed with the skill $d$ that the policy actually took. Our critic networks $Q(s,z,d)$ take the image $s$ and argument $z$ as input and have a $d$-headed output for each of the $d$ skills.

We do not use automatic KL tuning (standard in SAC implementations [57]) as we found it to be unstable; instead, we manually set entropy coefficients $\alpha_d$ and $\alpha_z$ for the policy (Equation (4)) and critic losses. In Kitchen, $\alpha_d = 0.1$, $\alpha_z = 0.01$; in LIBERO $\alpha_d = 0.1$, $\alpha_z = 0.1$. These values are obtained by performing a search over $\alpha_d = \{0.1, 0.01\}$ and $\alpha_z = \{0.1, 0.01\}$.

In FurnitureBench, we set $\alpha_z = 0.5$ and $\alpha_d = 2.0$ to prevent the policy losses from diverging significantly as we use RLPD [58] with a high critic update ratio of 2 per environment step and a higher policy update ratio of 2 per environment step.

## B.2 Baseline Implementation Details

**Oracle.** Our oracle baseline is RAPS [9]. We run RAPS to convergence and report final performance numbers because its expert-designed skills operate on a different control frequency; it takes hundreds of times more low-level actions per environment rollout. We only evaluated this method on Franka Kitchen as the authors did not evaluate on our other environments, and we found the implementation and tuning of their hand-designed primitives to work well on other environments to be non-trivial and difficult to make work.

**SPiRL.** We adapt SPiRL, implemented on top of SAC [57], to our image-based settings and environments using their existing code to ensure the best performance. For each environment, we tuned SPiRL parameters (entropy coefficient, automatic entropy tuning, network architecture, etc.) first and then built our method upon the final SPiRL network architecture to ensure the fairest comparison. SPiRL uses the exact same datasets as ours but without skill labels. We also experimented with changing the length of SPiRL action sequences, and similar to what was reported in Pertsch et al. [6], we found that a fixed length of 10 worked best. We also found fixed prior coefficients KL divergence to perform better with SPiRL for our environments than automatic KL tuning.

**EXTRACT-UVD.** Universal Value Decomposer (UVD) segments trajectories into sub-trajectories using VLM features for image goal-conditioned behavior cloning [63]. It was originally made for goal-conditioned imitation learning; we combine it with EXTRACT and adapt it for our setting of online reward-based reinforcement learning by using it to segment subtrajectories with the same VLM as EXTRACT, then treating each subtrajectory as a separate skill trajectory to condition EXTRACT's model on (without discrete skill extraction process). Essentially, this model acts as EXTRACT but with skill trajectories determined by UVD's trajectory segmentation method instead of that of EXTRACT. This also makes the comparison against our method more fair as it receives temporally extended skills, just like SPiRL or EXTRACT.

**BC.** We implement behavior cloning with network architectures similar to ours and using the same datasets. Our BC baseline learns an image-conditioned policy $\pi(a \mid s)$ that directly imitates single-step environment actions. We fine-tune pre-trained BC models for online RL with SAC [57].

**SAC.** We implement Soft-Actor Critic [57] directly operating on low-level environment actions with an identical architecture to the BC baseline. It does not pre-train on any data.

## B.3 Environment Implementation Details

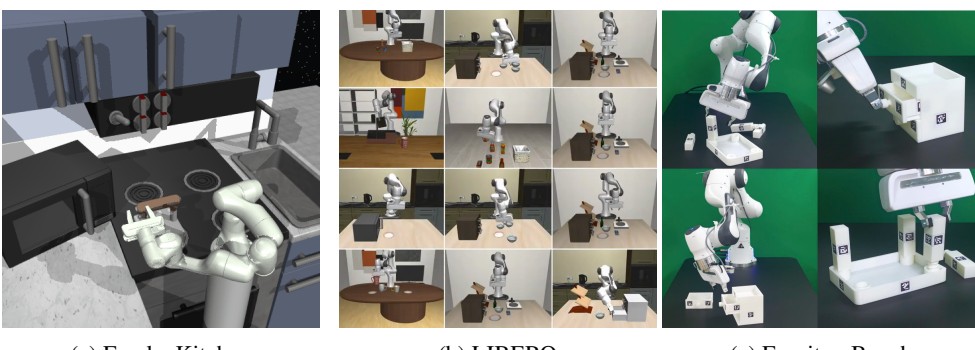

| (a) Franka Kitchen | (b) LIBERO | (c) FurnitureBench |

Figure 8: Our two image-based, continuous control robotic manipulation evaluation domains. **(a) Franka Kitchen:** The robot must learn to execute an unseen sequence of 4 sub-tasks in a row. **(b) LIBERO:** We evaluate 4 task suites of 10 tasks, each consisting of long-horizon, unseen tasks with new object, spatial, and goal transfer scenarios. **(c) FurnitureBench:** We evaluate online RL adaptation to unseen object and gripper placement randomizations.

**Franka Kitchen.** We use the Franka Kitchen environment from the D4RL benchmark [59] originally published by Gupta et al. [22] (see Figure 8a). The pre-training dataset comes from the "mixed" dataset in D4RL consisting of 601 human teleoperation trajectories each performing 4 subtasks in sequence in the environment (e.g., open the microwave). Our evaluation task comes from Pertsch et al. [6], where the agent has to perform an *unseen* sequence of 4 subtasks. The original dataset contains ground truth environment states and actions; we create an image-action dataset by resetting to ground truth states in the dataset and rendering the corresponding images. For all methods, we perform pre-training and RL with 64x64x3 RGB images and a framestack of 4. Sparse reward of 1 is given for each subtask, for a maximum return of 4. The agent outputs 7-dimensional joint velocity actions along with a 2-dimensional continuous gripper opening/closing action. Episodes have a maximum length of 280 timesteps.

**LIBERO.** LIBERO [60] is a continual learning benchmark built upon Robosuite [66] (see Figure 8b). For skill extraction and policy learning, we use the `agentview_rgb` 3rd-person camera view images provided by the LIBERO datasets and environment. For pre-training, we use the LIBERO-90 pre-training dataset consisting of 4500 demonstrations collected from 90 different environment-task combinations each with 50 demonstrations. We condition all methods on 84x84x3 RGB images with a framestack of 2 along with language instructions provided by LIBERO. We condition methods on language by embedding instructions with a pre-trained, frozen sentence embedding model [67], `all-MiniLM-L6-v2`, to a single $384$-dimensional embedding and then feeding it to the policy. For EXTRACT, we condition on language by conditioning all networks on language; $q, p_z, p_a, p_d$ are all additionally conditioned on the language embedding and thus the skill-based policy is also conditioned on language. We also condition all networks in all baselines on this language embedding in addition to their original inputs.

When performing additional fine-tuning to LIBERO-{10, Goal, Spatial, Object}, for all methods (except SAC) we use the given task-specific datasets each containing 50 demonstrations per task before then performing online RL. In LIBERO-{Goal, Spatial, Object}, sparse reward is provided upon successfully completing the task, so the maximum return is 1.0. In LIBERO-10, tasks are longer-horizon and consist of two subtasks, so we provide rewards at the end of each subtask for a maximum return of 2.0. Episodes have a max length of 300 timesteps.

**FurnitureBench.** FurnitureBench [61] is a real-world furniture assembly benchmark, where the task is to assemble 3D printed furniture pieces with a single Franka Arm (see Figure 8c). We closely reproduced the environment setup presented in the original paper through manual camera calibration of RealSense D435 cameras. For skill extraction and policy learning, we use two cameras, a wrist-mounted camera and a front mounted camera facing the arm workspace. To cluster skills, we embed both the wrist camera and front camera images with R3M and concatenate the embedding before clustering with K-Means ($K = 6$). Following the baselines implemented in the paper, we encode all RGB images with the frozen R3M video encoder to a 2048 dimensional vector first for all components of skill/policy learning. Also following the paper, we don't use any framestacking. Additionally, despite the presence of AprilTags, we do not use AprilTag-based state estimation for any part of our experiments; we perform purely image-based continuous control.

The pre-training dataset consists of 500 demonstrations from the `one-leg` assembly task collected by Heo et al. [61]. See Figure 9 for an image of the pieces used. The environment action space is absolute 3D position control plus a 6-dimensional rotation representation [68]. We run real-world online RL with RLPD [58], a sample-efficient actor-critic algorithm that uses a high critic update ratio, layer norms in the Q functions, a large number of Q functions, and samples from offline data and new online-collected data at a 50/50 ratio. We did not have to modify the policy/training objectives in Algorithm 4 from SAC for RLPD. To train EXTRACT's high-level policy with offline data in RLPD, we use the skill $d$ in the offline trajectories as high-level skill selection actions for $\pi(d \mid s)$ and encode offline sampled trajectory actions with the pre-trained, frozen encoder $q(z \mid \bar{a}, d)$ to obtain continuous arguments $z$ for $\pi(z \mid s, z)$. The offline data also comes with sub-task completion

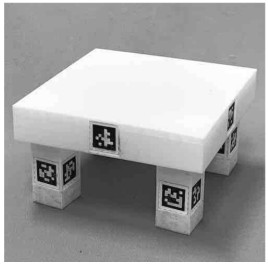

Figure 9: 3D-printed FurnitureBench table used for our `one leg` assembly task.

timestamps (picked up table top, placed it into the corner, picked up leg, inserted leg, screwed in) that we convert into $+1$ rewards for RLPD training for a maximum return of 5.

In our real-world RL setup, we run all methods for 100 trajectories with variations of 5cm for the initial object positions and end effector positions, along with $\pm 15$ degrees of initial end effector rotation. Meanwhile, we use a dataset of 500 demonstrations from the "low randomness" dataset split for `one-leg` assembly from [61], which contains no intentional randomness for both the object and end effector poses. This is challenging as the robot arm, camera positioning, etc., and now initial object and end effector locations, are all different from those in the dataset as collected by the FurnitureBench authors and the policy must *transfer* its knowledge to this new setting to solve the task. We provide two rewards when training online RL: $+1$ for completing an assembly sub-task successfully and $0$ for all other timesteps. The max return is also 5.

Episodes have a maximum length of 500 timesteps. Each trajectory takes approximately 50s of robot interaction time when run to completion and 30s to reset, resulting in $\sim$1.5 minutes of real-world time per trajectory.

## C  Additional Experiments and Qualitative Visualizations

In this section, we perform additional experiments and ablation studies. In Appendix C.1, we visualize 2D PCA plots of clusters generated by EXTRACT in all environments. In Appendix C.2, we analyze statistics of the skill distributions generated by EXTRACT. In Appendix C.3 for more ablation studies comparing using CLIP [23] or proprioceptive states instead of R3M [47] for clustering feature extraction. Finally, in Appendix C.4 we analyze skills extracted through UVD [63] against ours.

## C.1  Additional PCA Cluster Visualizations

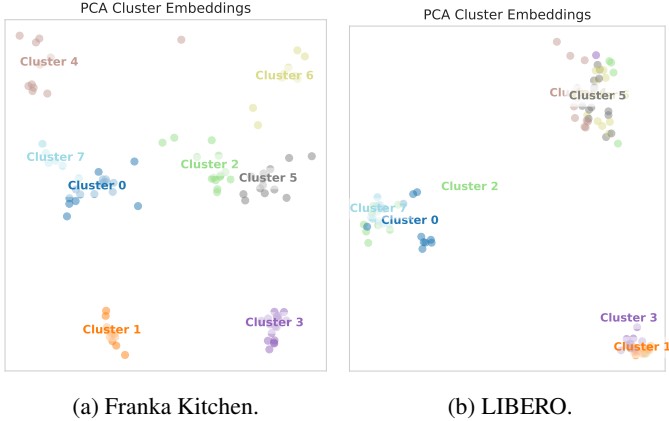

(a) Franka Kitchen.  (b) LIBERO.

Figure 10: 100 randomly sampled trajectories from environment pre-training datasets after being clustered into skills and visualized in 2D with PCA. Clusters are well-separated, even in just 2-dimensions with a linear transfromation.

Here we display PCA skill cluster visualizations in Figure 10. Franka Kitchen clusterings are very distinguishable, even in 2 dimensions. (this is the same embedding plot as in Figure 4 in the main paper). LIBERO-90 clusters still demonstrate clear separation, but are not as separable after being projected down to 2 dimensions (from 2048 original dimensions). However, in Figure 15 we clearly see distinguishable behaviors among different skills in LIBERO.

## C.2 Visualizing Cluster Statistics

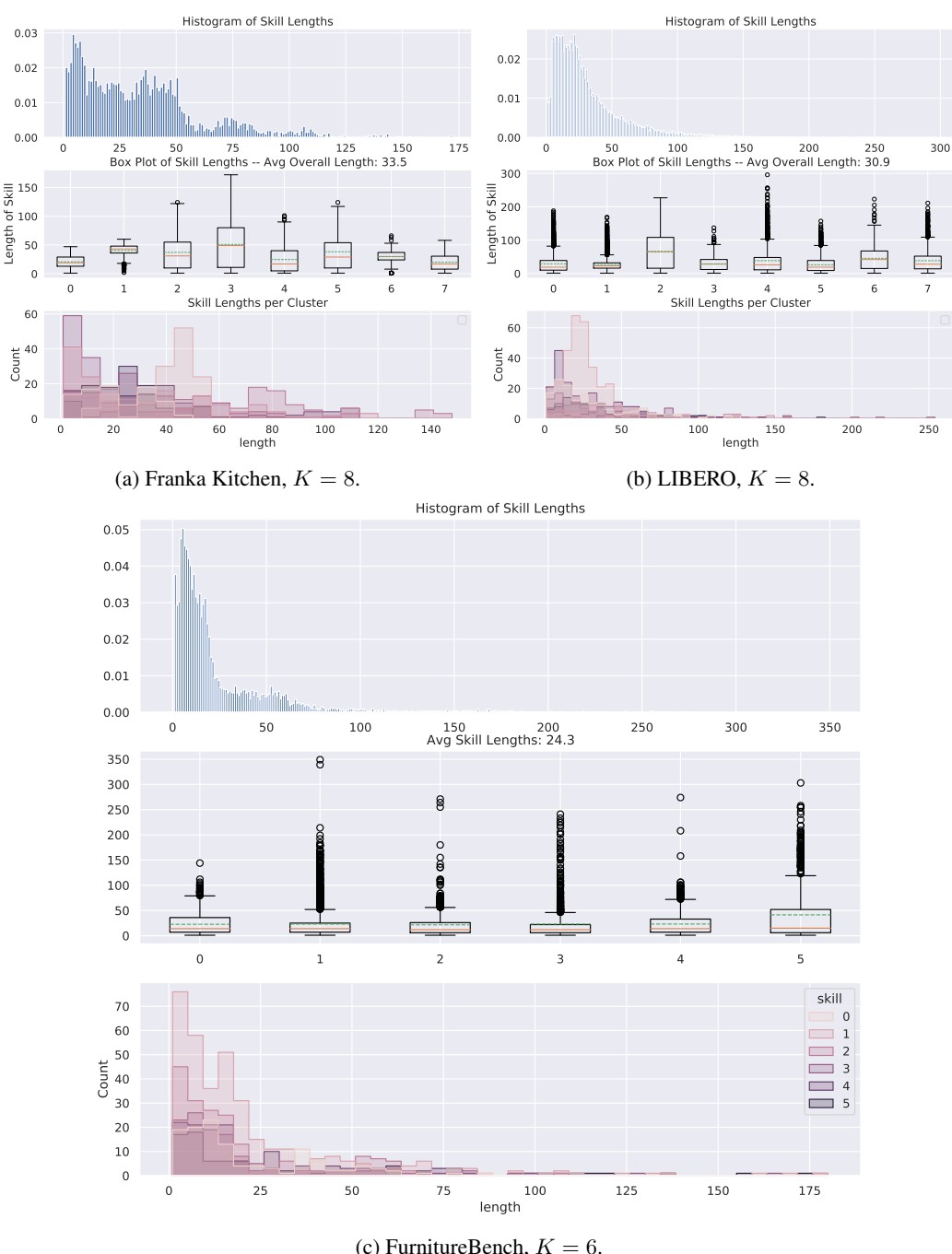

(a) Franka Kitchen, $K = 8$.

(b) LIBERO, $K = 8$.

(c) FurnitureBench, $K = 6$.

Figure 11: Skill/clustering statistics in all environments. We use the R3M VLM [47] and $K = 8$ for K-means. In FurnitureBench, $K = 6$. The top plots are skill length histograms for all skill trajectories combined, middle plots correspond to box-and-whisker plots with skill ID on the x-axis and lengths on the y-axis, and the bottom plots represent distributions of skill lengths separated by color for each skill ID.

We visualize skill clustering statistics in all pre-training environments in Figure 11. The plots demonstrate that average skill lengths are about 30 timesteps for all environments and that there

is clear separation among the different skills just in terms of the distributions of skill lengths that they cover. For a qualitative look at the skills, see Appendix D.

### C.3 Additional Ablation Studies

Here, we augment Section 5.4 with additional ablation studies. We compare against using **CLIP** [23] embeddings differences instead of R3M and against **Proprio**, i.e., robot joint and gripper state differences, on Franka Kitchen in Figure 12.

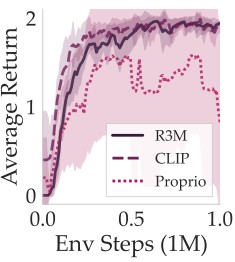

We originally chose R3M as the base VLM because, in contrast with R3M, CLIP is trained on images, not videos, on a general dataset of image-language pairs instead of a dataset of humans performing real-world tasks from an egocentric viewpoint that more closely mimics robotics environments (Ego4D, [69]). However, we can see that EXTRACT with CLIP performs on par with EXTRACT with R3M. This demonstrates that EXTRACT is robust to the choice of VLM for clustering; it's likely that CLIP was pre-trained on sufficient data to extract useful embedding differences for clustering. However, proprioceptive state differences are not as effective as proprioception can be difficult to directly obtain *high-level*, semantically meaningful skills from.

Figure 12: EXTRACT with R3M vs with CLIP or proprioceptive states.

### C.4 Visualizing UVD's Skill Extraction vs Ours

In Section 5.3, we found EXTRACT with UVD's skill extraction method to have unstable RL performance in Franka Kitchen and overall performed worse than EXTRACT with our skill extraction method.

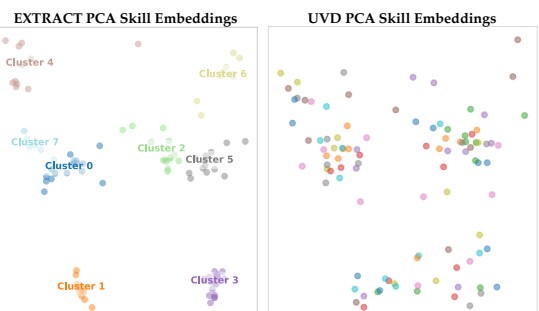

To analyze why, we plot R3M skill embeddings of skill trajectories extracted by EXTRACT against those of UVD, projected to 2 dimensions, in Franka Kitchen in Figure 13. We can see that UVD-generated trajectory embeddings are much harder to distinguish from each other than EXTRACT's, as evidenced by the distinct separation seen in the 4 corners of the plot compared to UVD.

Figure 13: 100 randomly sampled trajectories from Franka Kitchen after being projected to 2D with PCA. EXTRACT embeddings are identical to those in Figure 10.

This difference in trajectory separability, combined with EXTRACT's skill clustering approach that forms a discrete-continuous skill space for RL policies to learn new tasks with, helps explain the instability of RL with UVD in Franka Kitchen.

## D Visualizing skill trajectories

Here, we visualize skill trajectories in our environments. In Figure 14, we visualize purely randomly sampled clusters (i.e., without any cherry-picking) in Franka Kitchen, where we see skills are generally semantically aligned. For example, skill 3 trajectories correspond to manipulating knobs, skill 5 trajectories reach for the microwave door, and skill 7 trajectories are reaching for the cabinet handle.

We visualize LIBERO skills in Figure 15, where we can also see that skills are generally aligned.

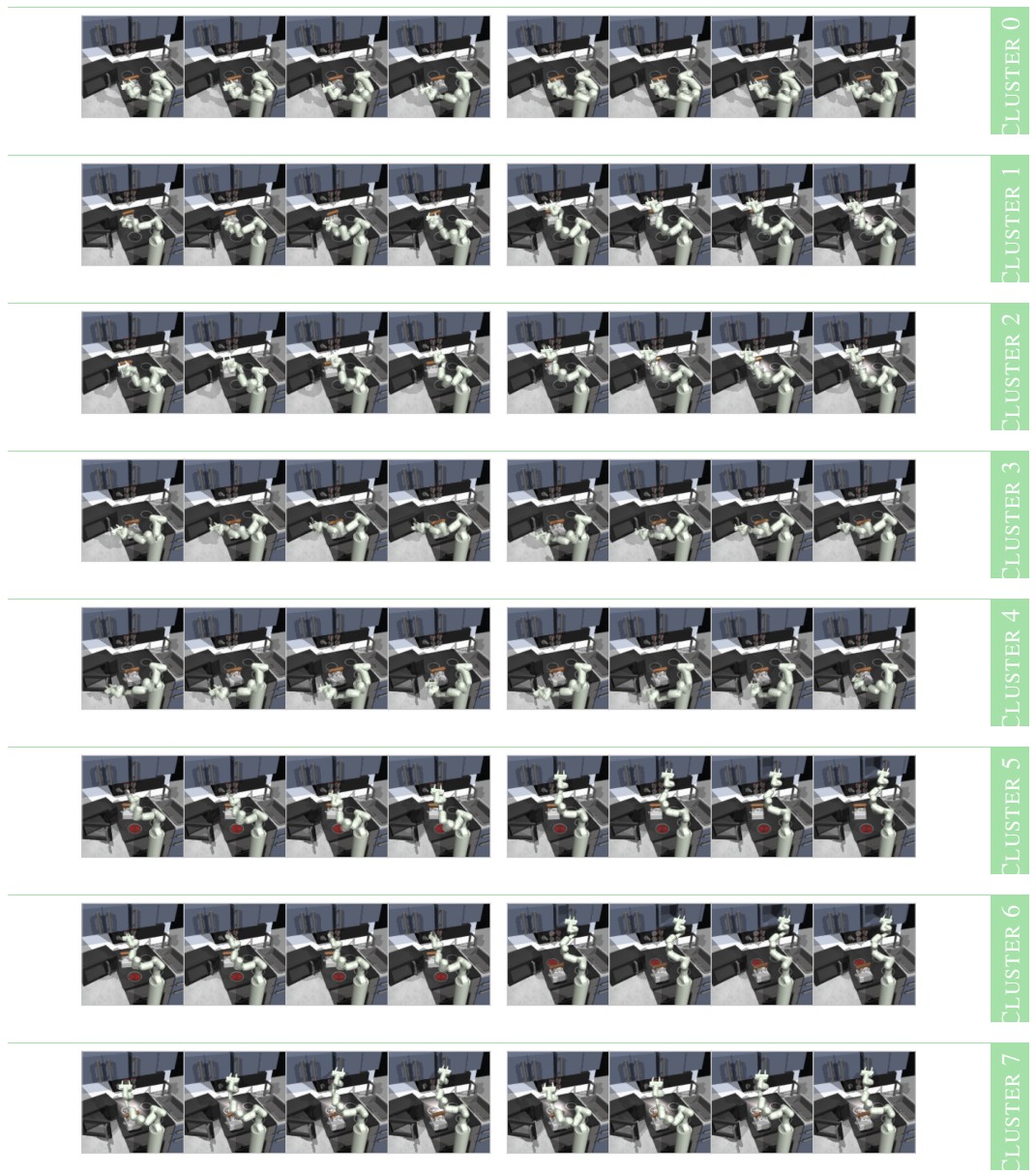

Figure 14: Kitchen skill visualizations. We randomly sample 2 labeled skill trajectories (no cherry-picking) and visualize the trajectory's images in sequence after labeling with EXTRACT's skill extraction phase. Clusters are generally *semantically aligned*.

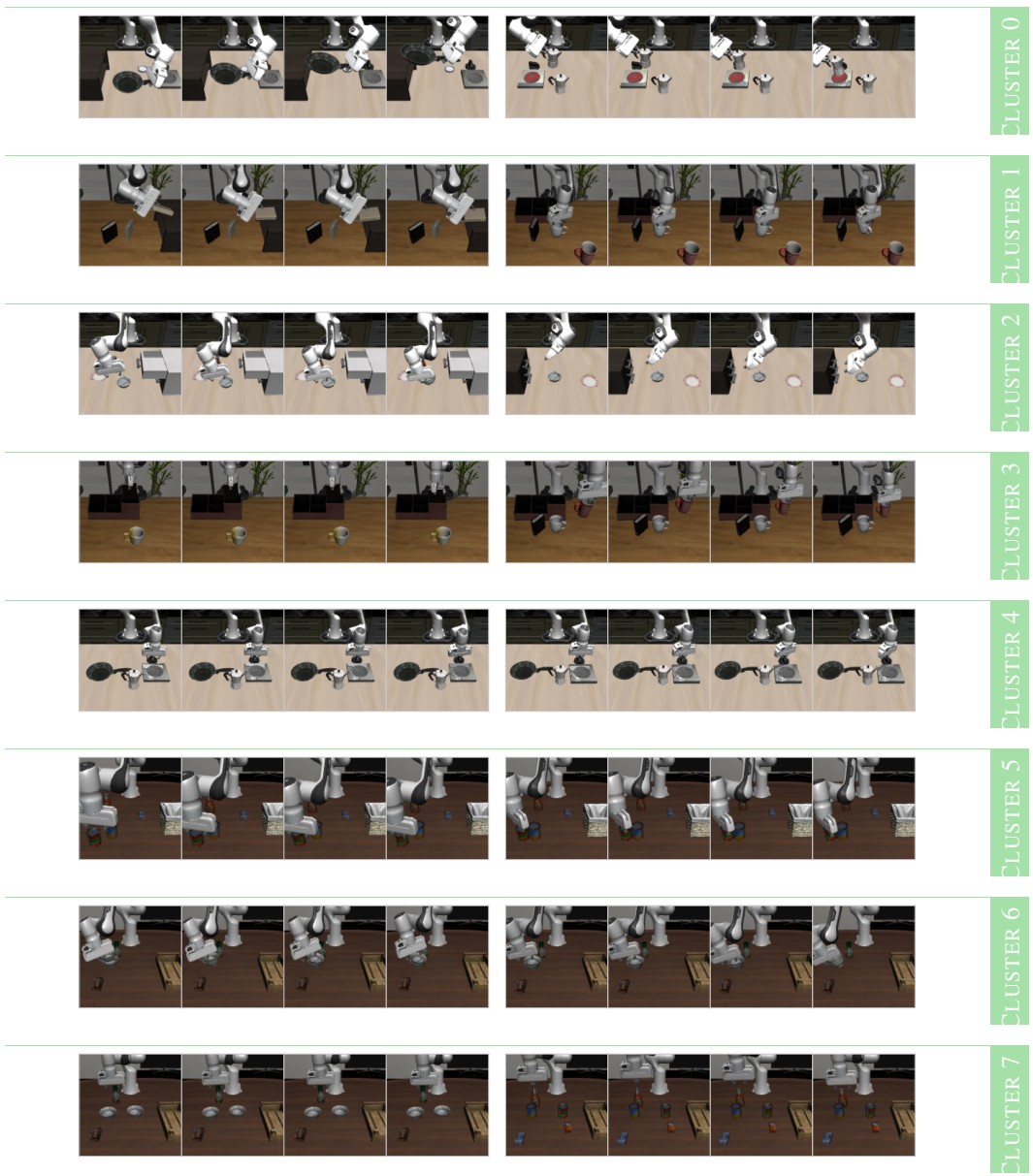

Figure 15: LIBERO. We randomly sample 2 labeled skill trajectories (no cherry-picking) and visualize the trajectory's images in sequence after labeling with EXTRACT's skill extraction phase. Clusters are generally *semantically aligned*.

# E   EXTRACT RL Performance Analysis

Our method's performance improvement over SPiRL is likely due to two reasons: longer average skills and a semantically structured skill-space instead of the random latent skills that SPiRL learns. In Section 5.3 we analyze the semantically structured skill-space. Here, we additionally analyze the longer average skills.

As plotted in Appendix Figure 11, EXTRACT extracts skills of various lengths, many of which are quite long. This translates into longer-executed skills: we plot a histogram of the lengths of the skills the skill-based policy actually learns to use at convergence in Franka Kitchen in Figure 16. EXTRACT-executed skills average 25 timesteps in length as compared to 10 for SPiRL. We experi-

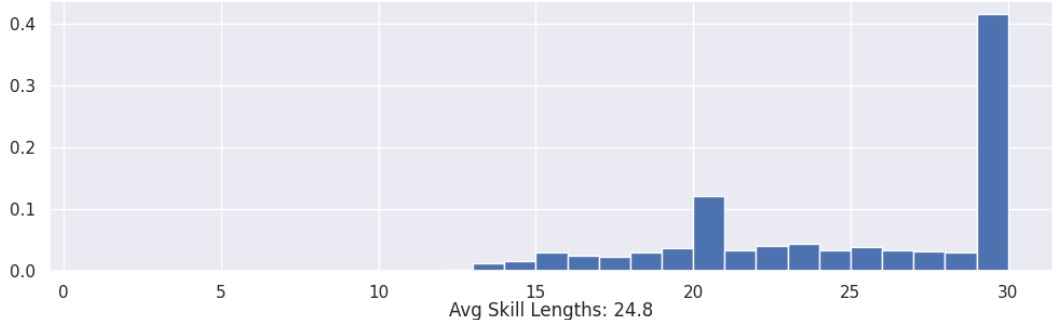

Figure 16: Skill lengths histogram of actually used EXTRACT skills in Franka Kitchen at training convergence. As explained in Appendix B.1, we limit skill execution lengths to 30 in Franka Kitchen.

mented with longer skill lengths for SPiRL, but online RL performance suffered, a finding consistent with results presented in their paper [6].

Longer skills shorten the effective time horizon of the task by a factor of the average skill length for the skill-based agent because the skill-based agent operates on an MDP where transitions are defined by the end of execution of a *skill* which can be comprised of many low-level environment actions. By shortening the task time horizon, the learning efficiency of temporal-difference learning RL algorithms [70] can be improved by, for example, reducing value function bootstrapping error accumulation as there are less timesteps between a sparse reward signal and the starting state.

## F  Limitations

While EXTRACT enables efficient transfer learning, we still need the initial dataset from environments similar to the target environments for learning skills from. It would be useful to extend EXTRACT to data from *other robots* or *other environments* significantly different from the target environment to ease the data collection burden—possibly wth sim to real techniques [71]. Furthermore, in future work, we plan to combine our method with offline RL [72, 73, 74, 75, 76, 77] to learn skills from suboptimal data *without* the need to interact with an environment, targeting even greater sample efficiency. Furthermore, while EXTRACT is more sample-efficient than all other comparisons, it still requires many online samples to learn to execute new tasks with RL. We plan to investigate future directions that will allow us to combine offline learning approaches such as offline reinforcement learning with skill learning on the offline dataset to allow even more efficient transfer. Finally, EXTRACT requires image observations for the VLMs; skill learning from more input modalities would be interesting future work. Processing more input modalities to learn skills would be interesting future work.

