# OpenReview forum: "EXTRACT: Efficient Policy Learning by Extracting Transferable Robot Skills from Offline Data"
_robot-learning.org/CoRL/2024/Conference — CoRL 2024_

### Official Review · Reviewer_Jq78 · 2024-07-19
**Interesting and well-presented method, but experiments are somewhat lacking**

**Originality:** 4
**Technical Quality:** 2
**Clarity Of Presentation:** 4
**Potential Impact:** 3
**Recommendation:** 3
**Confidence:** 4

**Review:**

Pros
+ The paper is clear and easy to understand, and the authors' approach is very interesting. Specifically, the use of pretrained visual language models to segment the trajectories and do skill alignment is novel and appears easy to reproduce, and the learning scheme in the offline skill learning phase was enlightening (the collection of losses in equation 2 was great). In addition, the math is well written and the notation is easy to follow. Overall, I like the approach and think it would be valuable for researchers working on constructing skill libraries for robots.
+ The figures are extremely helpful visuals for understanding the approach. I especially found Figure 2’s Offline Skill Learning figure helpful for understanding how the various submodules and training objectives related to each other.
+ I found the use of trajectory-level embedding differences to be interesting, as well as the inclusion of auto-regressively predicting the proportion of the skill completion in the decoder.
+ I appreciated the experimental validation in terms of the usage of two different domains (Franka Kitchen and LIBERO). I also appreciated the ablation studies, especially the one on how the number of k-means clusters impacted performance.

Cons
- There were no experiments done on robot hardware, which makes the evaluation relatively limited.
- While I feel the experiments did a good job of evaluating the offline skill learning and online skill-based RL phase, I don’t feel the experiments did a proper job of evaluating the offline skill extraction stage. It’s not clear to me how much better their approach is compared to standard non-machine learning skill extraction methods (such as segmenting based on changes in proprioceptive data or changes in gripper state), and they didn’t compare to other state-of-the-art methods for segmenting trajectories into skills like UVD [1]. Because of these lack of experiments, I can not properly evaluate how effective the skill segmentation approach is compared to others.
- The offline skill learning phase seems to only make use of the first state in the segmented trajectory, which seems like it may be wasting some potential state data that is available.
- The embeddings in the offline skill extraction appear to be calculated per-image, but this makes it seem like it would be hard to distinguish between skills that involve temporal information. For example, when opening / closing a door, it is not possible to discern from a single still image whether the robot is opening or closing the door, you would need additional frames from the past to determine which way the door is being manipulated. So it seems that this method would not be able to separate these kinds of skills into different clusters, even though they have very different effects.
- The color coding for “argument prior” and “selection prior” in Figure 2 is extremely hard to distinguish.
- In figure 4, besides cluster 4 in which the robot is opening the cabinet, it is not clear to me at all what is semantically meaningful about these clusters. It is possible the images are too small to discern properly, and it would help if the authors indicated what are the meaningfully changed parts of the images.

**Quality Of The Limitations Section:**

2

**Questions For Rebuttal:**

- Why were there no experiments done comparing the offline skill extraction method to other skill extraction methods like standard segmentation approaches based on proprioception / gripper state change, or other SOTA methods like UVD [1]?
- How would the offline skill segmentation approach deal with a skill like opening and closing, where each individual image can appear in both trajectories, and is only discernable when using multiple frames? Would it just make one cluster for opening/closing?
- In figure 2, should there be a reconstruction arrow pointing from [a1,a2,a3,...] to the skill decoder?
- Just to clarify, In equation 2, does z come from the skill arg encoder?
- Why was k-means chosen, which requires the number of clusters to be chosen ahead of time, instead of a clustering method that doesn’t require a hand-specified number of clusters like DBSCAN?

**Robotics Focus:**

3

**Summary Of Paper:**

This paper proposes EXTRACT, an approach to using pretrained visual-language models to segment and cluster trajectories into a discrete set of skills, and a novel framework for learning parameterized policies in an offline manner, as well as learning high-level skill selection policy in an online manner.

**Summary Of Recommendation:**

I lean towards accepting this paper, but the experimental validation is lacking (both in terms of comparison to baselines for skill extraction and hardware experiments)

---

### Official Review · Reviewer_2VAd · 2024-07-20
**nice paper, could benefit from better evaluation**

**Originality:** 3
**Technical Quality:** 3
**Clarity Of Presentation:** 4
**Potential Impact:** 3
**Recommendation:** 4
**Confidence:** 3

**Review:**

Pros:
1. The approach employs VLM to obtain latent state embedding which makes it robust to appearance variations, furthermore the embedding is made trajectory independent by subtracting features of initial state.
2. Median filtering helps in reducing the outliers samples which helps in learning useful tasks.
3. Autoregressive CVAE helps in learning variable length skills which is vital to learning diverse skills.
4. Entropy regularisation based hierarchical reinforcement learning based formulation ensures exploration which is step-up over imitation learning especially for transferring to new tasks.

Cons:
1. One of the major limitations is the setting that offline dataset and learning new skills should share the same environment. Therefore, the approach is not able to utilise demonstrations from different datasets.
2. Related issue comes from the assumption that demonstrations are meaningful, therefore robot is unable to explore and collect data, because then K-Meanes need large number of clusters that makes the skill conditioned policy learning inefficient as seen in ablation studies when number of clusters grow over 15.
3. Another issue is with offline phase of skill learning, which is never updated online thus transfer learning scope is limited.
4. The evaluation baselines ae rather sparse and do not compare with skill discovery+imitation learning baselines such as xskills [1].

[1] Xu, Mengda, et al. "Xskill: Cross embodiment skill discovery." Conference on Robot Learning. PMLR, 2023.

**Quality Of The Limitations Section:**

3

**Questions For Rebuttal:**

1.	Ablation study suggests that performance stays similar for 3-12 clusters, and only 3 skills are enough to solve most of the franka kitchen tasks. Can authors give insight on this issue?
2.	Training hierarchical policies could have stability issues, what additional measures were used to fine tune policies?

**Robotics Focus:**

3

**Summary Of Paper:**

This paper presents a unified approach towards skill discovery from an offline dataset and skill-guided hierarchical reinforcement learning. A pre-trained visual language model is used to extract robust embeddings from images. K-Means clustering is then employed to categorize these visual embeddings into a fixed number of skills. Following this, a stochastic action decoder is learned. This decoder samples variable-length feasible trajectories from discrete skill labels using a conditional variational autoencoder in an autoregressive manner. Finally, the approach is transferred to new tasks online. Two maximum entropy reinforcement policies are learned for selecting optimal skills and optimal actions given the skills. The effectiveness of this approach is demonstrated by its ability to transfer to new tasks on two datasets. This approach provides a comprehensive method for offline skill discovery and online task adaptation, showcasing the potential of combining unsupervised learning with reinforcement learning in complex environments.

**Summary Of Recommendation:**

Overall, this paper is presents a new formulation of employing VLM features for skill driven RL, the approach is well motivated and clearly presented. However, the evaluation is bit weak, the paper could benefit from comparison with stronger baselines and real robot experiments.

---

### Official Review · Reviewer_moSg · 2024-07-21

**Originality:** 4
**Technical Quality:** 4
**Clarity Of Presentation:** 5
**Potential Impact:** 3
**Recommendation:** 4
**Confidence:** 4

**Review:**

This paper is well written and clear. The novelty appears to be identifying skills from trajectories using VLM’s, providing a vector of parameters to discrete skills, and encoding skills of variable length. Skill-based learning with unsupervised skill extraction are ideas that can be built on following this work to lead towards more efficient and general policy learning. The main limitation is no mention of hardware experiments in future work, including discussion on what would need to be improved for this to work on a robot.

The skill decoder doesn’t consider state information, only skill ID and arguments. Is it expected the arguments are constant for a particular skill variant i.e. the skill decoder is executing open-loop behaviors?  Is the idea that the trajectory for the base skill will get the robot part of the way, then learning the arguments changes the original skill enough to solve the task? From these questions, how much could the test environment be varied from what is seen during training, and what are some strategies for getting this system to work on real hardware?

The decoder outputs the entire trajectory during online learning (Algorithm 4). How is this different during skill training, i.e how is the decoder trained autoregressively?

Have there been experiments on the choice of VLM for the extraction step, why was R3M chosen? Will this be the same, looking towards the extension mentioned of extracting skills from other robots and environments?

For Libero, the skill-based policy is also conditioned on language. What’s the effect of not having this conditioning? Shouldn't the skill index be enough?

**Quality Of The Limitations Section:**

3

**Questions For Rebuttal:**

See above.

**Robotics Focus:**

3

**Summary Of Paper:**

This paper shows that discrete sets of skills can be extracted with embedding differences from a trajectory of images using a VLM (R3M). With these discrete skill-sets, after smoothing and clustering, a skill decoder network can be trained to output actions given a skill ID and vector of arguments. Policy learning then becomes identifying what skill and arguments for a given task. This method is demonstrated on the Franka Kitchen and LIbero experimental setups in simulation. Results show improvement over previous method SPiRL.

**Summary Of Recommendation:**

This is a strong paper. A pathway to hardware experiments should be considered. -> Recommendation changed after rebuttal and inclusion of hardware experiments.

---

### Author Rebuttal · Authors · 2024-08-11

We thank all of the reviewers for their valuable feedback! We are glad that the reviewers find our paper on *efficient policy learning by extracting transferrable robot skills from offline data* to be well-written, novel, and very interesting.

Following meta-reviewer suggestions, we have run new experiments and included the following in the updated paper:

- **Real Robot Experiments**: Per moSg, 2VAd, and Jq78’s suggestions, we've added **real robot RL transfer learning experiments** on [FurnitureBench](https://clvrai.github.io/furniture-bench/) [1], a *furniture assembly benchmark*, where EXTRACT’s final performance outperforms the baseline by **1.6X** after 100 episodes of **real-world RL** (see Table 1, reproduced below).
| Method       | Start | End  |
|--------------|-------|------|
| SPiRL        | 1.35  | 1.55 |
| **EXTRACT** | **1.90** | **2.50** |
- **Additional Baselines**: As requested by 2VAd and Jq78, we've added a comparison with Universal Value Decomposer [2], outperforming it by up to **3x** (see Figure 5).
- **Additional Ablations**: We've ablated R3M for skill extraction against CLIP [3] and proprioception, showing EXTRACT’s robustness to VLM choice and outperforming proprioception (see Figure 12). (moSg, Jq78)
- **Generalization**: We clarified how EXTRACT generalizes to new environments, emphasizing that our existing LIBERO experiments test skill transfer to **40 unseen tasks**. (moSg, 2VAd)
- **Distinguishing Skills with Motion**: We clarified that EXTRACT can *already* distinguish skills defined by dynamic states. (Jq78)

**We have addressed all major concerns in the meta-review** with new experiments, clarifications, and writing changes. We address further individual reviewer concerns in per-reviewer responses.

**To summarize our contributions:**

- We introduce EXTRACT, an efficient transfer learning method for robotics that extracts a meaningful set of discrete, continuously parameterized skills from offline data for learning new tasks.
- We validate that EXTRACT learns well-aligned discrete skills.
- We demonstrate that EXTRACT agents transfer skills faster than prior methods across challenging, long-horizon, sparse-reward, image-based robotic tasks.

[1]: FurnitureBench: Real-World Furniture Assembly Benchmark, Minho Heo et al., RSS 2023.

---

### Decision · Program_Chairs · 2024-09-04

**Decision:**

Accept

**Comment:**

In summary, the initial reviews highlight the following strengths and weaknesses of this submission.

Strengths:
- The approach is novel and interesting.
- The paper is well written and clear.

Weaknesses:
- It is unclear how the method can distinguish skills which are defined by dynamic states.
- Generalization to variations of the environment unseen during training is unclear.
- Additional ablations, for instance, regarding the design choice for the VLM or conditioning on language are missing.
- Missing experimental comparison with baselines (e.g., xskills/UVD).
- The paper should evaluate on real robots or discuss how the approach could be scaled to real robots.

The author response and subsequent discussion addressed several of the reviewers' concerns including real robot experiments. Subsequently, all reviewers voted for accepting the paper. One reviewer asked for evaluation of alternative hierarchical RL methods on the extracted skill segments, which has not been addressed properly in the rebuttal and is recommended to be included in the final version of the manuscript.